# Hydroxyapatite and Titanium Dioxide Nanoparticles: Radiolabelling and In Vitro Stability of Prospective Theranostic Nanocarriers for ^223^Ra and ^99m^Tc

**DOI:** 10.3390/nano10091632

**Published:** 2020-08-20

**Authors:** Petra Suchánková, Ekaterina Kukleva, Eva Nykl, Pavel Nykl, Michal Sakmár, Martin Vlk, Ján Kozempel

**Affiliations:** Department of Nuclear Chemistry, Faculty of Nuclear Sciences and Physical Engineering, Czech Technical University in Prague, Břehová 7, 11519 Prague 1, Czech Republic; petra.suchankova@fjfi.cvut.cz (P.S.); ekaterina.kukleva@fjfi.cvut.cz (E.K.); emmalkova@gmail.com (E.N.); nyklpavel1@gmail.com (P.N.); michal.sakmar@fjfi.cvut.cz (M.S.); martin.vlk@fjfi.cvut.cz (M.V.)

**Keywords:** hydroxyapatite, titanium dioxide, nanoparticles, radium, ^223^Ra, technetium, ^99m^Tc, theranostic, radiolabelling, in vitro stability

## Abstract

Hydroxyapatite and titanium dioxide are widely used materials in a broad spectrum of branches. Due to their appropriate properties such as a large specific surface area, radiation stability or relatively low toxicity, they could be potentially used as nanocarriers for medicinal radionuclides for diagnostics and therapy. Two radiolabelling strategies of both nanomaterials were carried out by ^99m^Tc for diagnostic purposes and by ^223^Ra for therapeutic purposes. The first one was the radionuclide sorption on ready-made nanoparticles and the second one was direct radionuclide incorporation into the structure of the nanoparticles. Achieved labelling yields were higher than 94% in all cases. Afterwards, in vitro stability tests were carried out in several solutions: physiological saline, bovine blood plasma, bovine blood serum, 1% and 5% human albumin solutions. In vitro stability studies were performed as short-term (59 h for ^223^Ra and 31 h for ^99m^Tc) and long-term experiments (five half-lives of ^223^Ra, approx. 55 days). Both radiolabelled nanoparticles with ^99m^Tc have shown similar released activities (about 20%) in all solutions. The best results were obtained for ^223^Ra radiolabelled titanium dioxide nanoparticles, where overall released activities were under 6% for 59 h study in all matrices and under 3% for 55 days in a long-term perspective.

## 1. Introduction

The progress in the development of nanomaterial technology has a significant influence on all scientific and everyday life applications [1,2]. The massive expansion of nanomaterials is also seen in medicine, such as in bandages with antimicrobial nanosilver or with antibiotic capsules and nanoparticles (NPs), which are applicable for delivering drugs, light, heat, etc. [3]. In nuclear medicine, inorganic NPs could be used as one of the possible carriers for diagnostics or therapeutic radionuclides. At present, hydroxyapatite (HAp) [4], BaSO_4_-NPs [5], Ag-NPs [6], LaPO_4_-NPs [7], TiO_2_ [8], superparamagnetic iron oxide NPs [9] are under examination. An important advantage of nanoparticles is a large specific surface area and radiation stability which allows them to resorb ions and also to retain recoil radionuclides [10]. For these reasons, hydroxyapatite (*n*HAp) and titanium dioxide nanoparticles (*n*TiO_2_) were selected.

Hydroxyapatite is a natural material occurring in bones and teeth. Its artificial analogue is used in medicine as a part of bone and tooth implants [11,12]. Other HAp applications, for example as an additive to sunscreens [13] or biologically active material for cell proliferation and osteogenic differentiation, are under research [14]. The content of calcium and phosphorus in HAp structure in stoichiometric ratio of 1.67 (Ca to P) leads to the most stable modification. Sub-stoichiometric ratio is found in the natural HAp, so the calcium content is lower. However, HAp with calcium deficit is still stable from the biological point of view. Moreover, the natural structure of HAp is usually more complex and contains other ion traces, such as fluorine, etc. [12]. Precipitation from aquatic solutions is the easiest and the fastest method for HAp preparation and can be used on a large scale [15].

Titanium dioxide is commonly and widely used material in many consumer products. Due to its optical properties, TiO_2_ is also used as a functional part of sunscreens [16]. This material was selected due to its biocompatibility and relatively low toxicity [17,18,19,20]. Moreover, its preparation is fast, reliable and proper for large-scale production [15]. In nuclear medicine, TiO_2_ has found its place as a sorbent in a ^68^Ge/^68^Ga generator, where ^68^Ge is sorbed on titania and ^68^Ga is eluted by sterile ultra-pure 0.1 mol/L hydrochloric acid [21].

Hydroxyapatite was already labelled with diagnostic radionuclides e.g., ^99m^Tc, ^18^F. Radiolabelled NPs with ^99m^Tc were developed for bone cancer imaging [4]. Nanoparticles with citrate modified surface for ^18^F radiolabelling were prepared by Sandhöfer et al. [22]. The literature on therapeutic radionuclides ^177^Lu and ^223^Ra for *n*HAp radiolabelling is also available. For therapy of hepatocellular carcinoma [23] and the treatment of rheumatoid arthritis [24], ^177^Lu-HAp were studied. In all these studies, in vitro and in vivo preliminary studies were performed and the results were promising. Radium-223 was used for radiolabelling of NPs [25] and spherical HAp granules [26]. Preliminary experiments with HAp, which has programmable properties, were also performed with short-lived radionuclides of copper and zinc [27]. In a dosimetric study, HAp labelled with ^153^Sm and ^90^Y was compared in radiosynoviorthesis [28]. ^169^Er and ^177^Lu labelled HAps were used in radiation synovectomy [23,29]. Besides medicine, HAp is studied for radionuclide removal from radioactive waste [30].

Only a few publications are devoted to radiolabelling of *n*TiO_2,_ describing mostly the studies of titania biodistribution in tissues and organs. Radiolabelling with ^48^V was used for in vivo investigation of nanoparticles’ transport in lungs [31]. Vanadium-48 was also applied for the toxicological study of *n*TiO_2_ quantitative biokinetics and clearance in rats after intravenous injection, oral application and intratracheal instillation [17,18,19]. Short-term in vivo biodistribution studies were performed with ^18^F, where ^18^O-enriched TiO_2_ was irradiated by proton beam [32]. Another diagnostic radionuclide ^68^Ga as an emitter of Cerenkov radiation was used together with TiO_2_ for photodynamic therapy [33]. A different type of radiolabelling study was the low-temperature diffusion of titanium radioisotopes—^44^Ti, ^45^Ti [34]. Targeted alpha therapy using TiO_2_ was already studied by its ^225^Ac-radiolabelling [8] and Ag-dopped TiO_2_ particles by its ^211^At-radiolabelling [35]. A number of studies dealing with uranium and uranyl salts sorption on TiO_2_ describe its beneficial impact for long-term storage as an extra barrier for radionuclide sorption for radioactive waste repositories [36,37].

This work develops the already-published results by Kukleva et al. [15], Suchánková et al. [38] and Suchánková et al. [39]. The main interest in the current paper was given to ^223^Ra labelling due to its therapeutic properties. The total released energy of ^223^Ra in a form of alpha particles through all the decays is high enough to destroy cell DNA without broad damage to surrounding tissue. Radium-223 is already used as a Xofigo^®^ (RaCl_2_) approved by EMA and FDA [40,41] for palliative treatment of bone metastases of prostate cancer. Due to its having similar pathways to Ca^2+^, Ra^2+^ is targeted into bones, where it replaces calcium. Simultaneously, such an elegant natural targeting is a limitation for Xofigo^®^, which cannot be targeted to other tissues. This disadvantage could be solved by an appropriate carrier. Selected NPs could serve as targeting vector for ^223^Ra together with its daughter radionuclides. As a complementary radionuclide for theranostic approach, ^99m^Tc, as the most frequently used diagnostic radionuclide in medicine, was selected. Among its benefits belong suitable energy of gamma and suitable half-life, moreover, it is easily obtained from ^99^Mo–^99m^Tc generator [42]. Radium-223 generators are also under investigation.

This work aims to verify the radiolabelled nanoparticles and to provide evidence as to whether they can be used as radionuclide carriers in nuclear medicine. The work was mainly focused on the NPs radiolabelling and the behaviour and stability of the labelled NPs in different media. Firstly a suitable and fast radiolabelling strategy with appropriate yield was developed. For this purpose, two strategies of radiolabelling were selected. The first strategy was the radionuclide sorption on ready-made NPs and the second one was the intrinsic labelling, where the radionuclide was incorporated into the structure of NPs at the stage of their preparation. The second task was to determine the influence of biologically relevant media on the in vitro stability of the radionuclide-NP carrier system. For this in vitro study physiological saline, bovine blood plasma and serum, 1% and 5% albumin solution were used. These data will allow to determine if the nanocarriers could be used for further in vivo experiments, and if the radionuclides remain with the carriers and the radionuclides are not released to the surrounding tissues.

## 2. Materials and Methods

All chemicals were of analytical grade and were used without further purification: tetrabutyl orthotitanate (TBOT), propane-2-ol (IPA), ammonium hydroxide, calcium nitrate tetrahydrate, sodium chloride, diammonium hydrogen phosphate, and sodium azide purchased from Merck (Darmstadt, Germany); SnCl_2_ purchased from The British drug houses Ltd. (Poole, UK), and bovine plasma and serum, albumin lyophilised (pH 7) purchased from Biowest (Nuaillé, France). Demineralized water of 18 MΩ/cm^−1^ was obtained from Millipore (Burlington, MA, USA) water purification system.

Gamma spectra were measured with a HPGe detector and analysed using the Maestro Software (ORTEC, Oak Ridge, TN, USA). Overall activities were measured with a well-type NaI(Tl) crystal CII CRC-55tW (CAPINTEC, Ramsey, NJ, USA). For mixing of samples, Stuart SSM3 rocker (Cole-Parmer Ltd., Vernon Hills, IL, USA) was used and separation was made on VWR Micro Star 12 centrifuge (VWR International, Radnor, PA, USA). All experiments were performed under aseptic conditions in laminar box Airflow 150 UV (Esi FLUFRANCE, Arcueil, France).

### 2.1. Preparation of Nanomaterials

The detailed description of both hydroxyapatite and titanium dioxide nanoparticles’ synthesis and characterization was already published by Kukleva et al. [15]. Here, only synthesis is described briefly. The basic characteristic of prepared nanoparticles are summarised in Table 1.

For the synthesis of *n*HAp, 1.2 M Ca(NO_3_)_2_ in demineralized water was used. For correct synthesis running, pH was set and maintained at 11 by ammonium hydroxide solution. Afterwards, the same volume of 0.7 M (NH_4_)_2_HPO_4_ was dropwise added and mixed overnight. Then, prepared nanoparticles were washed with demineralized water (3×) and dried under vacuum.

The selected preparation method of *n*TiO_2_ was the hydrolysis of TBOT. The mixture of TBOT and IPA (1:4, respectively) was dropwise added into demineralized water in ultrasonic generator and the solution was mixed at the laboratory temperature. After that, prepared nanoparticles were washed with physiological saline (3×) and IPA (1×) and dried under vacuum.

### 2.2. Preparation of ^223^Ra and ^99m^Tc Stock Solutions

The stock ^223^Ra solution was obtained from ^227^Ac/^227^Th/^223^Ra generator prepared at the Department of Nuclear Chemistry at our laboratory according to Guseva et al. [43]. The column was filled with anion-exchanger Dowex-1 × 8 resin. The elution of ^223^Ra was performed by 0.7 M HNO_3_ in 80% methanol. Gamma-spectrometric analysis was used for radionuclide purity detection of ^223^Ra(NO_3_)_2_ solutions (breakthrough of parents’ radionuclides in the eluate). The eluted ^223^Ra(NO_3_)_2_ solutions were dried and reconstituted in demineralised water [25].

The ^99m^Tc solution stock was gained from commercial generator DRYTEC^TM^ (GE Healthcare LTD, Chicago, IL, USA). The elution was performed by physiological saline.

### 2.3. Radiolabelling Procedure

Two strategies of radiolabelling with both ^223^Ra and ^99m^Tc were selected. The first strategy was the surface radiolabelling, where the radionuclide was sorbed on the ready-made NPs’ surface. The previously prepared NPs (5 mg) were dispersed in 300 µL of physiological saline for ^223^Ra radiolabelling or in 500 µL of fresh stannous chloride solution (480 mg/L) for ^99m^Tc radiolabelling. Consequently, ^223^Ra or ^99m^Tc solutions were added. The radioactivity of ^223^Ra was added from 5 to 10 kBq and the radioactivity of ^99m^Tc solution was from 60 to 100 MBq. The samples were mixed for one hour at laboratory temperature and then washed with physiological saline (3×).

The second strategy was the intrinsic labelling, where the radionuclide was incorporated directly into the NPs’ structure. Radiolabelled *n*HAp were prepared in the following way: 35 μL of 1.2 M Ca(NO_3_)_2_ was added to 500 μL of demineralized water, then pH was set to 11 by 1 M ammonium hydroxide solution. Then, the ^223^Ra solution was added in a small volume so that the activity added was in the same range as for surface labelling. Finally, 35 μL of 0.7 M (NH_4_)_2_HPO_4_ was added and the mixture was stirred for 1 h at laboratory temperature. In the case of ^99m^Tc radiolabelling, the procedure was basically the same, but instead of ^223^Ra solution, ^99m^Tc was added. The samples were washed with physiological saline (3×).

Radiolabelled *n*TiO_2_ was prepared in the following way: tetrabutyl orthotitanate in IPA (1:4, 70 μL) was dropwise hydrolysed in physiological saline (300 µL) with already-added ^223^Ra solution or in stannous chloride solution (500 µL) with added ^99m^Tc. The radioactivity was in a similar range to in the previous case. The samples were mixed for 1 h at laboratory temperature and washed with physiological saline (3×).

All experiments for both radiolabelling strategies for both materials were repeated six times and resulting yields were calculated.

### 2.4. In Vitro Stability Studies of Radiolabelled Materials

In vitro stability studies were performed in several matrices: physiological saline, bovine blood plasma and serum and human albumin solutions (1% and 5%). Nanoparticles were centrifuged after labelling and washing, and the solution was replaced with a new matrix. All samples were performed in triplets. The samples were incubated at laboratory temperature and shaken during the incubation. The matrices were replaced with the same but fresh solution after 2–7–12–17–26–35–59 h from the begging of the experiment (in the case of ^99m^Tc last interval was removed and the previous one was shortened from 9 to 5 h due to its shorter half-life). Samples and supernatants were measured on NaI(Tl) scintillation detector. The percentages of released radioactivity were specified separately for ^223^Ra-NPs (short-term) and ^99m^Tc-NPs. Long-term stability studies were provided for ^223^Ra, where the matrices were replaced every 11 days (T_1/2_(^223^Ra) = 11.4 d) five times (total 55 days).

## 3. Results and Discussion

### 3.1. Radiolabelling

Two strategies of NP radiolabelling were tested: surface (S) and intrinsic (I) labelling. The radiolabelling yields were determined according to Equation (1)
(1)$$\%Y\;=\;\frac{A_{NPs}}{A_{init}}\;\times \;100\%$$
where *A_NPs_* is the final activity of radiolabelled nanoparticles and *A_init_* is an initial activity. The longest radionuclide of the decay chain, the ^223^Ra, was measured. Daughter radionuclides were not specifically determined. All radiolabelling yields for both strategies and materials were higher than 94% (Table 2).

Some data about HAp radiolabelling were already published by other scientists. Spherical hydroxyapatite with diameter 900–1000 μm (10 mg or 19 spherical granules) were incubated 24 h with ^223^Ra. The radiolabelling yield was approximately 80% [26]. Another paper focused on hydroxyapatite was by Albernaz et al. [4]. Hydroxyapatite powder (<210 μm) was radiolabelled with ^99m^Tc (approx. 3.7 MBq). Particles were incubated for 10 min and the average radiolabelling yield was 98.5%. These results are in good agreement with obtained experimental data.

To date, there has been no literature found dealing with ^223^Ra radiolabelling of titanium dioxide. The presented results of *n*TiO_2_ radiolabelling yields were slightly higher than *n*HAp yields and strategy of intrinsic labelling showed the same or even better yields than surface radiolabelling. However, the difference was statistically undetectable. It is important to note that all the results correspond with earlier published data from sorption and kinetic studies [38,39].

Both labelling strategies could be considered for radiopharmaceuticals preparation. Each type, with its advantages, could be appropriate for another utilisation. On the one hand, surface radiolabelling is preferable for the often-prepared radiopharmaceutical doses. Standardised kits with ready-made NPs of defined size can be prepared with required sterile and apyrogenic properties easily. The kit type radiopharmaceuticals are, at present, often used in nuclear medicine departments and could be stored and used in laboratory conditions. On the other hand, the intrinsic labelling may be used as rapid method, where both the nanoparticle preparation and the radiolabelling procedures are performed in a single step. This method could be useful for a variety of modifications of radiolabelled nanoparticles, which cannot be provided before radiolabelling.

### 3.2. In Vitro Stability Studies of Radiolabelled Nanomaterials

In vitro experiments were designed as pseudo-open systems, where used liquid was replaced in defined periods under free air conditions. All samples were shaken during incubation period. These in vitro stability tests deal with the radionuclide-carrier system only. The system was considered as stable if the radionuclide release from the material to the media was observed minimally. The colloidal stability of the nanoparticles themselves was not studied, however this important parameter requires further research, particularly for in vivo applications.

Both radiolabelled nanomaterials were studied in vitro in the following solutions: physiological saline, bovine blood plasma and bovine blood serum, 1% and 5% human albumin solution. The percentage of released activity, %*A*, was performed according to Equation (2)
(2)$$\%A\;=\;\frac{A_{sup}}{A_{sup}+A_{NPs}}\;\times \;100\%$$
where *A_sup_* is activity of supernatant and *A_NPs_* is activity of labelled separated nanoparticles.

Physiological saline is the most frequently used solution in medicine, therefore it was selected as a comparative standard solution. The blood proteins serve as transporters of various compounds, therefore, two different solutions of albumin were also used for examination. Further, it is necessary to study labelled nanoparticles’ behaviour in plasma and serum. The plasma composition consists mostly of water and proteins such as albumins, globulins and fibrinogen, then ions, saccharides etc. The difference between blood plasma and serum is the presence of fibrinogen and clotting factors in plasma [44]. Commercially available plasma also contains sodium citrate as a stabilizing agent.

Hydroxyapatite nanoparticles in figures are given always as (a) and (b) and *n*TiO_2_ as (c) and (d). The first radiolabelling strategy, the surface labelling (S), is shown as (a) and (c) and the second one strategy, the intrinsic labelling (I), as (b) and (d). Error bars were not included in in vitro stability figures due to better clarity and ranged under 0.5%.

The short-term in vitro stability study of ^223^Ra radiolabelled nanoparticles is shown in Figure 1. The difference between the two studied nanomaterials can be seen. In the case of *n*HAp, the total released activity was around 60% in saline and bovine blood plasma after 59 h from radiolabelling. The released activities in saline were unexpectedly high. However, it can be possibly explained by the lower stability of *n*HAp and its dissolution at lower pH or by the influence of relatively high concentration of the sodium ions (0.9% sodium chloride solution).

Despite this fact, short-term stability experiments in saline showed that labelled nanoparticles could be temporarily stored in saline before the injection. Plasma experiments showed high released activities in comparison with serum results, which can be caused by proteins or sodium citrate interference. To eliminate the sodium citrate influence stability, studies with 1% solution were performed, but they did not show such a massive effect. It is important to note that the concentration of sodium citrate in plasma was unknown and was not provided by the manufacturer. Thus the influence of this well-known complexation agent cannot be excluded. The short-term stability of ^223^Ra-*n*HAp in serum had a very promising performance.

The released activity from ^223^Ra-*n*TiO_2_ was lower than 6% for surface radiolabelling and lower than 2.5% for radionuclide incorporation. In opposite to *n*HAp, the lowest released activity was in saline and then in bovine blood serum for both radiolabelling strategies. The released activity in the rest of the biological matrices was slightly higher but the differences were negligible compared to statistical deviations. It could also be noticed that in the case of surface labelling, the worst results were obtained again in plasma. Another difference compared to *n*HAp is the slight contrast between the radiolabelling strategies. In the case of intrinsic labelling, the released activities are slightly lower than in the surface labelling.

Total released activities in long-term in vitro studies (Figure 2) were similar to short-term studies, which can be explained by distribution coefficients and rather fast kinetics, where the equilibrium was reached before matrix replacement. Comparing the short-term and the long-term studies it could be seen that the ion-exchange and the resorption of radionuclides back on the nanoparticle surfaces play an important role. This effect could cause in vivo radionuclides’ release in the case of NPs depo accumulation, where the radionuclides are retained in the tissue.

Despite similar values of the released activity (from 10 to 50%), ^223^Ra-*n*HAp showed better stability in saline in the long-term than in the short-term study. The highest effect on the activity release had the albumin solution, which showed similar results for all four series of stability studies with ^223^Ra-*n*HAp. Considering the serum experiments with total released activity of about 10%, it can be concluded, that the serum does not have such a negative impact on the radiolabelled materials, therefore *n*HAp still can be taken as a suitable material, however, this requires further studies with other radionuclides, a modified labelling strategy or the carrier modifications.

Released activities from *n*TiO_2_ were lower than 3%, which may be caused by radionuclide resorption during the longer studied intervals. The lowest value was detected in the reference saline solution as in the previous case. Again, the differences among matrices for ^223^Ra-*n*TiO_2_ were negligible. In the case of *n*TiO_2,_ the effect of lower activity release for intrinsic labelling was observable.

To support the theranostic idea of inorganic nanoparticles as carriers, short-term in vitro stability study of ^99m^Tc radiolabelled nanoparticles was performed and the results are summarized in Figure 3. The behaviour of ^99m^Tc-labelled nanoparticles was different from the ^223^Ra-labelled ones. Contrary to ^223^Ra, the released activities of ^99m^Tc were lower in the case of *n*HAp and higher for *n*TiO_2_.

In the case of ^99m^Tc-*n*HAp, the released activities were under 20% for both radiolabelling strategies in 31 h. However, there was high released activity in blood plasma, which was around 60%. This could be caused by a sodium citrate present as a preservative agent. For better visualisation of obtained results, plasma data were not included in Figure 3 in the case of *n*HAp due to fast and high activity release, where it was higher than 50% after only 10 h from labelling. The lowest released activities were found in 5% albumin solution for both type of radiolabelling. It is also the enormous difference against the ^223^Ra-*n*HAp. This difference between in vitro stability studies of both studied radionuclides may be caused especially by their dissimilarity, e.g., valence, reactivity and other properties originating from type of metal, where the radium is alkaline earth metal and technetium is transition meal.

The stability of ^99m^Tc-HAp was also studied by Albernaz et al. [4]. However, the stability experiments were performed only in reaction solutions containing stannous chloride in 24 h. The average overall released activity was 6%. This is in good agreement with obtained experimental results from short-term stability studies of ^99m^Tc-*n*HAp.

The highest released activities from *n*TiO_2_ were around 15% in 5% albumin solution in the case of surface radiolabelling and even around 25% in bovine blood plasma in the case of intrinsic labelling. In all samples with titania, the lowest released activities were in physiological saline.

Our results showed that the sorption and kinetic experiments with ^99m^Tc on *n*HAp and *n*TiO_2_ are necessary. They could bring the answers about mechanism of technetium uptake and identify the rate-controlling sorption process as it could have an influence on the in vitro stability of ^99m^Tc-NPs. In the case of ^223^Ra, these experiments were already performed (see previous publications) and it was found that ion exchange mechanism plays the main role in sorption on *n*HAp and *n*TiO_2_.

In general, it is possible to conclude that the overall released activities were lower for ^223^Ra-*n*TiO_2_, than for any other studied material. The best performance was shown by *n*TiO_2_ with ^223^Ra, however, overall released activities from ^99m^Tc radiolabelled nanoparticles were similar for both types of materials and radiolabelling strategies. Although the released activities from ^223^Ra-*n*HAp were relatively high, it is still a promising carrier according to serum stability results. Previously published results [15,38,39] showed that both materials are suitable as carriers and both radiolabelling strategies showed high radiolabelling yields for inorganic nanomaterials.

The nanocarriers preparation and radiolabelling is the first step in new radiopharmaceutical development. Consequently, it could be necessary to upgrade the carrier to prepare core-shell structured particles or modify their surface with targeting and protecting substances, e.g., polyethylene glycol [6]. While the first step should prevent the radionuclide and its daughters from escape from the NPs, the latter should provide active targeting and better in vivo biodistribution. This concept could be also useful for *n*HAp modification to improve the behaviour of the radiolabelled nanoparticles in biologically relevant media.

## 4. Conclusions

Radiolabelling of nano-sized HAp and TiO_2_ was studied with ^99m^Tc or ^223^Ra using two labelling strategies, which were the labelling of ready-made particles (surface labelling) and radionuclides’ incorporation into the structure of the nanomaterial (intrinsic labelling). Both methods showed high labelling yields (>94%) for both nanoparticle types and both radionuclides. Consequently, the in vitro stability studies of radiolabelled nanoparticles were performed in biologically relevant media: physiological saline, bovine blood plasma and serum and 1% and 5% human albumin solutions. The most stable in all media were the ^223^Ra radiolabelled *n*TiO_2_, where the overall released activities in short-term stability experiments were under 6% in 59 h. Good results also were shown by both ^99m^Tc radiolabelled nanoparticles, where the overall released activities were under 20% in short-term aspect. In general, the worst stability shown by all materials was in plasma. From the long-term perspective, both types of NP showed good results, in some cases even better than from the short-term perspective. This may be caused by the resorption of radionuclides due to longer study periods. This, in fact, leads us to the conclusion that the stability experiments under static conditions—as usually performed and reported in the literature—may lead to false-positive results, that can be later compromised under the dynamic conditions, e.g., in animal in vivo models (see Figure 1 and Figure 2).

Both labelling strategies can find their application in practice. Despite the several high released activities, overall activity release in all experiments was stable and both nanomaterials are still promising radionuclide carriers and surface modification, for example, could improve the stability of radiolabelled nanomaterials. Based on the obtained results, it could be concluded that *n*HAp is more appropriate for local application and controlled activity release and *n*TiO_2_ is suitable for system application due to its good long-term stability.

## Figures and Tables

**Figure 1 nanomaterials-10-01632-f001:**
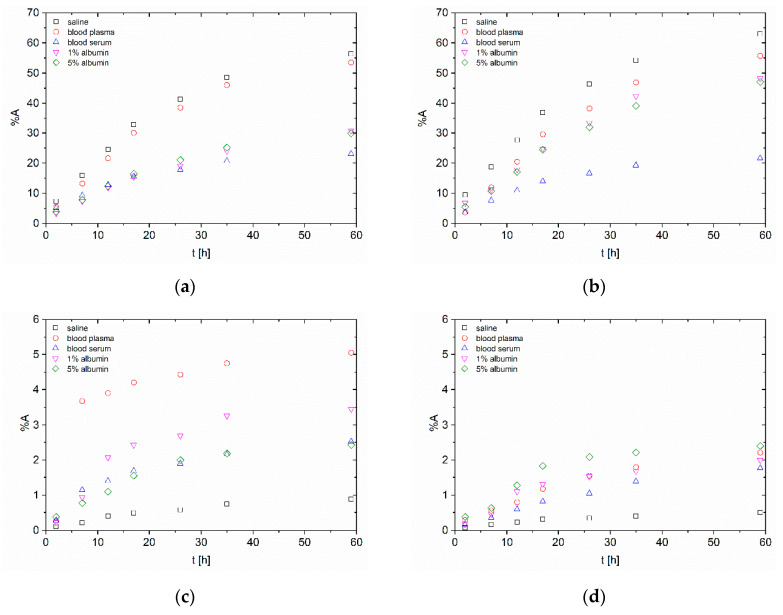
Short-term in vitro stability study of ^223^Ra-labelled nanoparticles (**a**) surface radiolabelling of *n*HAp; (**b**) intrinsic radiolabelling of *n*HAp; (**c**) surface radiolabelling of *n*TiO_2_; (**d**) intrinsic radiolabelling of *n*TiO_2_.

**Figure 2 nanomaterials-10-01632-f002:**
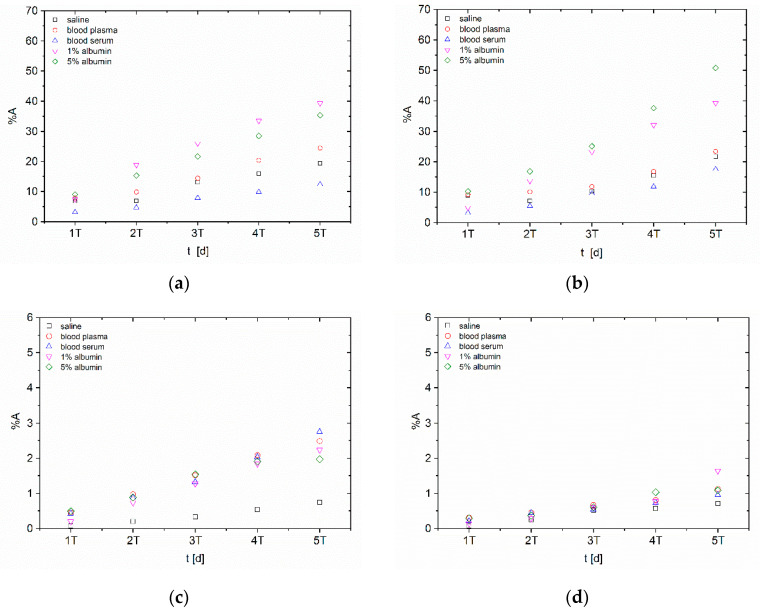
Long-term in vitro stability of ^223^Ra-labelled nanoparticles over approx. ^223^Ra half-life (T = 11 d) (**a**) surface radiolabelling of *n*HAp; (**b**) intrinsic radiolabelling of *n*HAp; (**c**) surface radiolabelling of *n*TiO_2_; (**d**) intrinsic radiolabelling of *n*TiO_2_.

**Figure 3 nanomaterials-10-01632-f003:**
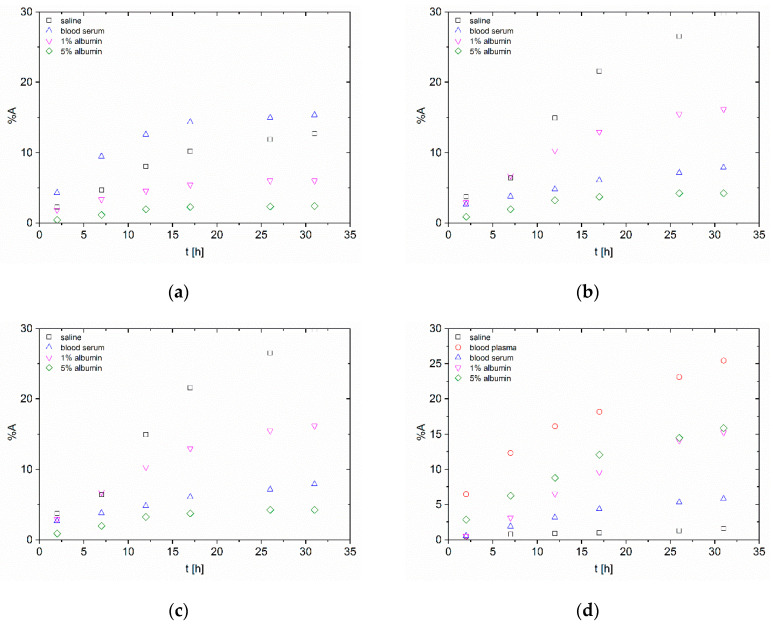
Short-term in vitro stability study of ^99m^Tc-labelled nanoparticles (**a**) surface radiolabelling of *n*HAp (except plasma results); (**b**) intrinsic radiolabelling of *n*HAp (except plasma results); (**c**) surface radiolabelling of *n*TiO_2_; (**d**) intrinsic radiolabelling of *n*TiO_2_.

**Table 1 nanomaterials-10-01632-t001:** Characteristics of *n*HAp and *n*TiO_2_ [15,38,39].

Characteristic	Unit	*n*HAp	*n*TiO_2_
Crystallite size	nm	5.18	2.64
Equivalent diameter	nm	21.7 ± 6.9	5.3 ± 1.7
pH applicability	pH	5–10	2–10
Specific surface area	m^2^·kg^−1^	117 ± 9	330 ± 10
Surface edge sites	mol·kg^−1^	5.10 ± 1.20	0.20 ± 0.01
Surface layer sites	mol·kg^−1^	0.15 ± 0.01	0.67 ± 0.01
Diffusion coefficient	cm^2^·min^−1^	2.50 × 10^−12^ ± 1.80 × 10^−12^	1.60 × 10^−14^ ± 0.96 × 10^−14^
Half-life of sorption	min	0.75 ± 0.18	0.51 ± 0.32

**Table 2 nanomaterials-10-01632-t002:** Yields for ^223^Ra and ^99m^Tc labelling of *n*HAp and *n*TiO_2_ (*n* = 6).

Labelling	^223^Ra [%]		^99m^Tc [%]	
	*n*HAp	*n*TiO_2_	*n*HAp	*n*TiO_2_
S	94.2 ± 0.5	98.7 ± 0.5	95.9 ± 1.5	98.4 ± 0.5
I	97.0 ± 0.5	99.1 ± 0.3	94.6 ± 0.4	97.6 ± 0.7

S—surface labelling, I—intrinsic labelling

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
