# Peer review of "Hydroxyapatite and Titanium Dioxide Nanoparticles: Radiolabelling and In Vitro Stability of Prospective Theranostic Nanocarriers for 223Ra and 99mTc"

_nanomaterials, 2020, doi:10.3390/nano10091632_

Round 1
Reviewer 1 Report
The manuscript “Radiolabelled Hydroxyapatite and Titanium Dioxide Nanoparticles: Preparation and In Vitro Stability of Prospective Theranostic Nanocarriers for 223Ra and 99mTc” investigates the potential of hydroxyapatite and titanium dioxide nanoparticles as nanocarriers of radionuclides which is of high importance in nuclear medicine. The paper is clearly written but there is information that is missing and could help in improving the clarity of presentation and discussion.
Firstly, the description of hydroxyapatite and titanium dioxide nanoparticles is missing. The synthesis and characterization of both nanomaterials were previously published (ref. 15). However, it would be good to present the basic characteristic of these materials similar to table 2. in ref. 39, either in the materials and methods section or in SI.
Secondly, there is no information on colloidal stability of investigated, non-labelled nanomaterials in investigated media. In the paper stability connected with released activity is presented. However, information of changes in the size and zeta potential of nanomaterials (non-labelled) in different media can help to elucidate their impact on observed “radioactivity” stability. Also determining if the nanomaterials are colloidally stable in a certain medium or if they aggregate, is crucial for designing applicable formulation of nanocarrier.
Therefore, I recommend this paper to be published after information on physico-chemical characteristics and colloidal stability of investigated nanomaterials are added.
Reviewer 2 Report
Dear authors, dear editor,
please find below my detailed comments for this manuscript.
Best regards
The manuscript submitted by Jan Kozempel and his co-workers provides insights in the investigation of stability of radiolabeled titanium dioxide and hydroxyapatite nanoparticles (NPs). The syntheses of the NPs themselves are not subject of this recent work program. Two methods for radiolabeling with 99mTc and 223Ra are presented, based on the principal of either surface sorption or matrix incorporation, which also were not developed in this recent work. The radiolabeled NPs were further incubated with saline (as reference) and several biological media (plasma, serum, 1% and 5% human albumin solutions) to examine the radiometal release over time. The release was determined using a centrifugation method and comparing the measured radioactivity in the pellet vs. the supernatant.
The authors are strongly focusing nanomaterial-based approaches in their research regarding targeted alpha therapy. There is only few known about TiO2 NPs and the 223Ra results seem to be promising from the first point of view, whereby these NPs are of high interest for future TAT developments. The stability of both matrices labeled with 99mTc is moderate.
The research field-related introduction of the draft is well written. The manuscript in general is logically structured but from the beginning of the aim of the study followed by methods and results/discussion section, the central idea is sometimes hard to catch. The authors provide good first results regarding their radiolabeled NP’s stability in biological media. Nevertheless, there are more biological experiments on cellular level needed to verify a potential use in nuclear medicine.
To complete the manuscript and the experimental setup I would strongly recommend to consider the following aspects for revising the experimental setup und suggest a resubmission afterwards.
Experimental improvement suggestions:
- Explain at least something about size distribution and morphology of the NPs. Are they small enough for intravenous application and what clearance behavior do you expect? Have you performed some initial in vivo experiments?
- Have you tested the NPs regarding protein corona formation? Do you observe macroscopic precipitation? Have you determined surface charge of the NPs?
- Also focus on testing the therapeutic potential of 223Ra-NPs in cellular assays. Since this is the third publication which is related to synthesis and characterization of these two types of radiolabeled NPs it is in my eyes needed to provide more cellular data. For this purpose, it would be necessary to design some reactive site on your NPs and to follow a standard targeting approach to easily examine specific/unspecific binding and in case of 223Ra-NPs also therapeutic effects on a cellular level. In the end you can proof that you universal carrier system for radiometals might be suitable for nuclear medicine applications.
- If you succeed within your in vitro studies, in vivo experiments would be of high interest to check, whether the NP’ body distribution and clearance behavior – since unwanted and unspecific binding is one of the largest limitations for future applications.
Further remarks:
- Line 15: specify “appropriate” properties
- Line 86: Xofigo® is not only used in trials, but EMA and FDA approved.
- Line 93: Do you mean specific generators? What is the relevance in the context? Set a reference here.
- Lines 145-147: Using 5-10 kBq of 223Ra in 5 mg of NPs seems to be too little. 10 kBq is one dose for an in vivo test, but approx. 50-150 µg are mostly injected. So in my mind you should either use a lower amount of NP or a higher amount of activity. This is also important to check before with your stability measurements, since the alpha recoil could strongly influence the overall stability.
- Line 155: A more commonly used term would be “room temperature”
- Line 202: “suitable for nuclear medicine” – in my eyes this is too far away, since there is not a ready-made NP yet, which means that the production process is not yet finished
- Chapter 3.2.: It is totally understandable to change the relevant time periods for the investigation of the stability of both NP types. For 99mTc a significantly shorter period is needed due to the shorter half-life. Where I have difficulties is the presentation of “short-term” and “long-term” data for 223Ra-NPs. These presented differences are not explainable at all. At least the overlapping time points (Fig 1 vs. 2) should be the same if I understood the setup correctly. To avoid misunderstanding I would suggest to leave the “short-term” data and only concentrate on the time points that are relevant for 223
- Conclusions should be chapter 4, not chapter 5
Round 2
Reviewer 1 Report
The authors have answered all the questions.
Author Response
Thank you again for your comments and suggestions.
Reviewer 2 Report
Dear Editor, dear authors,
I understand the reasons, why the authors cannot improve their manuscript by doing additional experiments. The minor remarks were revised. That is fine for me.
The synthesis and characterization of the described NPs is already shown in two previous papers, but the title of this manuscript says “preparation”. They added a brief preparation part as the reviewers suggested.
Nevertheless, some questions occur:
For me it is still not clear why different results were found for short time stability testing compared to long time testing. In the beginning of long time testing there should be the same values. For instance, why is the activity release of 223Ra below 20% (Figure 2B) in the long-term stability and higher than 40% (except in blood serum) for nearly the same time point in the short-term testing? What should be the same in my eyes. What is the reason? Can the authors comment this?
In conclusion, the manuscript contains a very detailed description of differently radiolabeled NP stability testing. I would recommend to accept this manuscript in the current form.
Author Response
Firstly, thank you for your comments and suggestions that helped to clarifiy and improve the paper.
We have modified the title to:
Hydroxyapatite and Titanium Dioxide Nanoparticles: Radiolabelling and In Vitro Stability of Prospective Theranostic Nanocarriers for 223Ra and 99mTc.
so it better corresponds to the manuscript´s content.
We have also clarified the difference in the short and long term experiments directly in the manuscript text.
The short-time and long-time stability tests were two different experiments. At the beginning there was always fresh sample of radiolabelled NPs. The difference between the two datasets was very likely due to more frequent exchange of the wash-solution matrix in short-time tests as it was mentioned in the manuscript. Both experiments are necessary for different points of view. The short-time stability experiments should predict the behaviour of radiolabelled NPs under dynamic conditions and thus faster surrounding solution exchange (e.g. in a blood stream). In the long-time stability experiments however, there is more time to achieve the equilibrium conditions and the probability that the radionuclide undergoes resorption on the NPs is thus higher, so the overall released activities could be lower compared to short term intervals. The same trend could be seen for both HAp and TiO2 NPs.
Further, the intervals and released activities shouldn't be directly compared in short and long-time experiments. It could be possible, if these datasets are provided as the results of one experiment under the same conditions. However, to do so, it is more challenging, considering the time, the radioactivity amount and the obtained results that would suffer from higher experimental and statistical deviations since the cumulative released activities for both experiments should be calculated to evaluate their performance. So therefore it is necessary to perform the short and the long-time stability tests as separate experiments and compare the overall gained data (not the single data points).
This, in fact, leads us to a conclusion that in general the stability experiments under static conditions - as usually performed and reported in the literature may lead to false-positive results, that can be later compromised under dynamic conditions (e.g. in animal in vivo models).